🔓 | **Open Peer Review** | Antimicrobial Chemotherapy | Research Article

# Dronedarone hydrochloride targets cardiolipin and phosphatidylglycerol to increase colistin susceptibility in gram-negative pathogens

Zhiying Liu,[1] Moyun Liu,[1] Zichu Wang,[1] Chenxiao Jiang,[1] Jianfeng Wang,[1] Xuming Deng,[1] Hongtao Liu,[1] Yanhong Deng,[1] Jiazhang Qiu[1]

**ABSTRACT**  Treating multidrug-resistant (MDR) infections has become progressively dependent on limited therapeutic options, particularly polymyxins, such as colistin. This reliance has precipitated a concerning epidemiological trend: the emergence and global propagation of plasmid-mediated (*mcr*) as well as chromosome-mediated polymyxin resistance. Consequently, escalating resistance rates will certainly lead to diminished clinical efficacy of colistin, correlating with elevated mortality in septic patients who already face therapeutic limitations. Utilizing antimicrobial potentiators to restore the sensitivity of resistant pathogens to polymyxins represents a promising pharmacological strategy for reinvigorating the clinical utility of these agents. Here, we demonstrate that dronedarone hydrochloride (DH) exhibits significant synergistic bactericidal activity with colistin against colistin-resistant strains. DH enhances the antibacterial potency of colistin by approximately 32-fold (MIC from 8 µg/mL to 0.25 µg/mL in ExPEC ECQ001), effectively reversing resistance phenotypes. *In vivo* therapeutic efficacy studies demonstrated that combination therapy achieved a statistically significant reduction in bacterial burden compared to colistin therapy alone. Mechanistic studies revealed that DH has the capacity for specific molecular interactions with two critical phospholipid components: cardiolipin and phosphatidylglycerol (PG) in bacterial membranes. This binding induces membrane disruption, impairs energy production, and stimulates oxidative stress, which collectively augment the bactericidal activity of colistin. These findings position DH as a viable antibiotic adjuvant with translational potential for combination therapies against MDR pathogens. The dual targeting of membrane integrity and redox homeostasis presents a strategic advantage in circumventing conventional resistance mechanisms, thereby extending the application potential of colistin in contemporary antimicrobial regimens.

**IMPORTANCE**  Colistin remains a last resort antibiotic for treating infections caused by extensively drug-resistant pathogens. However, the emergence of colistin resistance has significantly compromised its clinical utility. Our research identifies and characterizes that dronedarone hydrochloride (DH) restores bacterial sensitivity to colistin by binding to cardiolipin (CL) and phosphatidylglycerol (PG). Mechanistic studies revealed that DH bound specifically to CL and PG, thereby enhancing membrane disruption, impairing energy production, and stimulating oxidative stress levels, which collectively augment the bactericidal activity of colistin. These findings present DH as a lead compound for combating colistin resistance, while offering novel mechanistic insights into its role as a colistin potentiator.

**KEYWORDS**  colistin, dronedarone hydrochloride, antibiotic adjuvant, antimicrobial resistance

**Peer Reviewer** Kui Zhu, College of Veterinary Medicine, China Agricultural University, Beijing, China

Address correspondence to Jiazhang Qiu, qiujz@jlu.edu.cn, Yanhong Deng, dyh@jlu.edu.cn, or Hongtao Liu, liuht2018@jlu.edu.cn.

Zhiying Liu and Moyun Liu contributed equally to this article. The author order was determined by seniority.

The authors declare no conflict of interest.

See the funding table on p. 13.

Antimicrobial resistance (AMR) has evolved into a global public health emergency, driven by long-term antimicrobial overutilization and now escalating to critical levels (1). The proliferation of drug-resistant pathogens, particularly multidrug-resistant (MDR) bacteria, poses formidable challenges across public health, animal husbandry and veterinary medicine, environmental health, and other fields. Epidemiological projections underscore the severity of this crisis: A 2021 meta-analysis attributed 1.14 million fatalities directly to bacterial infections, with mortality rates predicted to approach 2 million annually by 2050 under current resistance trajectories. The associated economic burden was anticipated to surpass trillions of dollars (2). Of particular concern is the rise of extraintestinal pathogenic *Escherichia coli* (ExPEC), an opportunistic pathogen demonstrating serotype-specific virulence mechanisms that evade cross-protective immunity in both human and animal hosts (3, 4). The clinical impact of ExPEC infections is compounded by escalating resistance profiles among circulating strains. A surveillance study by Priyanka et al. quantified this threat by systematically screening 1,780 phytogenic samples, isolating 77 ExPEC strains (15% prevalence). Notably, over 75% of isolates exhibited multidrug resistance to 11 antibiotics, including β-lactams (cefoxitin and ceftazidime), folate pathway inhibitors (trimethoprim), and aminoglycosides (gentamicin). Most alarmingly, 100% of these phytogenic ExPEC strains demonstrated phenotypic resistance to colistin (5). This resistance profile substantially restricts therapeutic options, exacerbating morbidity risks in immunocompromised populations.

Polymyxins, a class of cationic polypeptide antibiotics produced by *Bacillus polymyxa* and ubiquitously distributed in terrestrial ecosystems, are structurally categorized into five distinct variants: polymyxin A, B, C, D, and E (6). Among these, polymyxin B and colistin (polymyxin E) have garnered significant clinical attention. However, concerns regarding dose-limiting nephrotoxic and neurotoxic adverse effects, along with the emergence of alternative antibiotic options such as aminoglycosides and fluoroquinolones, led to a marked decline in polymyxin usage during the 1970s (7). In recent years, the alarming rise of carbapenem-resistant gram-negative bacteria has necessitated the reintroduction of colistin as a critical therapeutic option for severe infections caused by MDR gram-negative pathogens (8). This resurgence has also coincided with an increase in colistin resistance, mediated primarily by chromosomal mutations and plasmid-encoded resistance determinants such as the *mcr* gene family, posing a formidable challenge to both human and veterinary practice (9). Alarmingly, the rapid evolution and global dissemination of MDR gram-negative bacteria threaten to exhaust our arsenal of effective antimicrobial agents, underscoring an urgent need for novel therapeutic strategies (10).

The principal mechanisms of resistance to colistin involve an array of multifaceted bacterial adaptations, such as structural modification of lipopolysaccharide (LPS) constituents, activation of barrier systems or broad-spectrum efflux pumps, expression of drug-degrading enzymes, and the heterogeneous resistance exhibited by bacterial populations (11, 12). The common resistance mechanism involves covalent modification of lipid A phosphate groups via 4-amino-4-deoxy-L-arabinose (Ara4N) and/or phosphoethanolamine (pEtN) moieties. This biochemical remodeling reduces the net negative charge of the bacterial outer membrane, thereby diminishing electrostatic interactions with the cationic amphipathic structure of colistin and impeding its initial membrane penetration (11). Efflux systems play a crucial role in colistin resistance across different species. For instance, the AcrAB-TolC pump in *Escherichia coli* (*E. coli*), the KpnEF system in *Klebsiella pneumoniae*, and the MexXY-OprM complex in *Pseudomonas aeruginosa* have been implicated in this phenomenon (13, 14). These macromolecular assemblies function as proton motive force-dependent extrusion pumps, actively reducing intracellular colistin accumulation through energy-dependent efflux while maintaining membrane integrity (13–15). The operational redundancy of these systems across species underscores their evolutionary conservation as a pan-bacterial resistance strategy (15).

Combination therapy with colistin has emerged as a promising strategy to enhance its clinical utility and mitigate resistance development (10). This pharmacological synergy operates through pleiotropic mechanisms targeting distinct bacterial vulnerabilities. For example, oxyclozanide disrupted bacterial membranes, increased the permeability, and susceptibility to colistin (16). Small molecule *mcr-1* antagonists (e.g., nordihydroguaiaretic acid) suppress lipid A remodeling by inhibiting phosphoethanolamine transferase activity, thereby preserving the target affinity of colistin (17). Compounds that elevate oxidative stress levels within bacteria can also potentiate the antibacterial activity of colistin (18). SLAP-S25 demonstrates broad-spectrum synergy by disrupting LPS biogenesis and targeting peptidoglycan (PG), fundamentally destabilizing cell architecture (19). Natural flavonoids have the capability to disrupt bacterial iron homeostasis by converting ferric iron to ferrous form. The accumulation of intracellular ferrous iron subsequently alters the bacterial membrane charge by interfering with the two-component *pmrA/pmrB* system, thereby promoting colistin binding and subsequent membrane damage (20). Short-chain fatty acids enhanced the antibacterial activity of colistin through multiple pathways, including inhibition of LPS modification and increased membrane damage (21).

Dronedarone hydrochloride (DH) is an amiodarone derivative used primarily to reduce arrhythmia recurrence, decrease ventricular rate, and prevent hospitalizations in patients with nonpermanent atrial fibrillation. Here, we found that DH specifically interacts with cardiolipin (CL) and phosphatidylglycerol (PG), destabilizing the bacterial membrane and increasing bacterial sensitivity to colistin. Moreover, it impairs proton motive force (PMF), decreases energy production, and induces oxidative stress in a dose-dependent manner. Importantly, DH significantly improved the therapeutic efficacy of colistin in a mouse infection model. Collectively, this work highlights the potential of DH as a potent colistin adjuvant, underscoring the importance of exploring novel combination therapies to address the growing challenge of antibiotic resistance. In brief, understanding the complex mechanisms of resistance and identifying effective adjunctive agents like DH are critical steps toward developing innovative treatment paradigms against MDR pathogens.

## RESULTS

### DH is a potent colistin adjuvant

Colistin remains an essential therapeutic option for treating infections caused by carbapenem-resistant gram-negative bacteria. However, the resistance to colistin is continuously severe, which seriously affects its clinical application. Consequently, our study aimed to identify compounds from the FDA-approved drug library that could potentiate the activity of colistin. Through MIC testing, we identified compound DH (purity = 99.77%) as a potential colistin synergist (Fig. S2C). Checkerboard assays confirmed the superior synergistic effect between DH and colistin against *E. coli* ECQ001 (Fig. 1A, FICI = 0.05 ± 0.00). To determine the synergism between DH and colistin, growth curve and synergistic bactericidal assays were carried out on *E. coli* ECQ001. In the monotherapy assay, neither DH nor colistin was sufficient to eliminate the bacteria, while their combination resulted in complete bacterial eradication within 24 h. (Fig. 1B and C). Importantly, the addition of DH mitigated the development of colistin resistance, as evidenced by lower increases in MIC values (twofold) when compared to colistin monotherapy (fourfold) (Fig. 1F).

Meanwhile, we found that DH also synergized with colistin against *E. coli* DH5α-pME6032 and *E. coli* DH5α-pME6032 + *mcr-1* (Fig. 1D and E), and the FICI was very close. This suggested that the synergy of DH and colistin was not dependent on the presence of *mcr-1*. To examine the universality of this combination against various bacteria, we subsequently evaluated the FICI of three other gram-negative bacterial species. DH potentiated the activity of colistin against these strains regardless of pre-existing colistin resistance (Fig. 2A through C; Table 1). And the cooperative bactericidal action confirmed the broad-spectrum activity of the DH-colistin combination (Fig. 2D through F). These

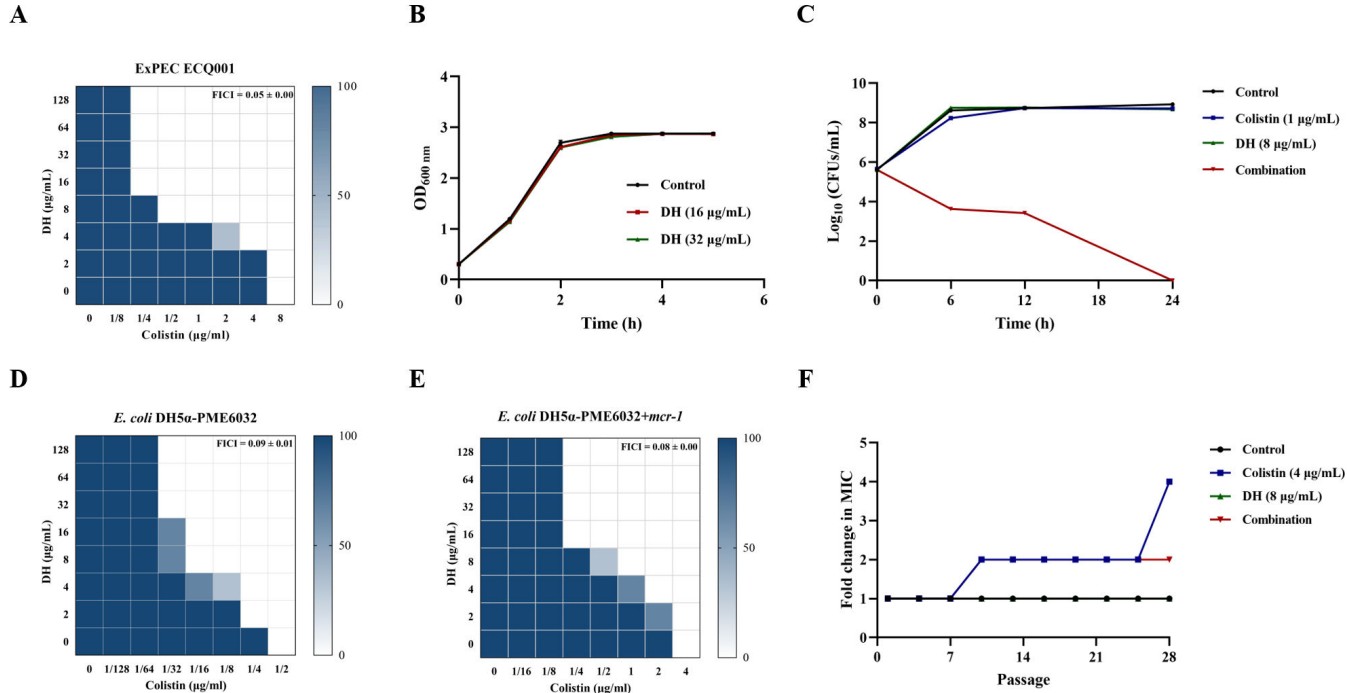

**FIG 1** Synergistic antimicrobial effects of DH and colistin. (A) Representative checkerboard assay of DH and colistin against *E. coli* ECQ001. (B) Growth curve of *E. coli* ECQ001 in different concentrations of DH. (C) Time-dependent killing curve of *E. coli* ECQ001 treated with DH, colistin, and their combination. Representative checkerboard assay of DH and colistin against (D) *E. coli* DH5α-PME6032 and (E) *E. coli* DH5α-PME6032 + *mcr-1*. (F) Resistance development of *E. coli* ECQ001 following serial passaging for 28 days in the presence of DH, colistin, or the combination.

findings indicated that the combination of DH and colistin was an effective method to exterminate the gram-negative bacteria and diminish the potential emergence of resistance.

## DH promotes membrane permeability and oxidative damage

Colistin, a cationic polypeptide antibiotic, exerts its bactericidal activity by interacting electrostatically with the anionic phosphate groups of lipid A, a component of the LPS layer that constitutes the outer leaflet of the gram-negative bacterial outer membrane. This interaction disrupts the integrity and permeability of the outer membrane, ultimately resulting in cell death. Given that the observed synergism between DH and colistin is consistent across various gram-negative bacteria. It suggested that DH might operate a common mechanism underlying this enhanced antibacterial effect. Based on this observation, we hypothesized that the addition of DH might augment the disruptive impact of colistin on the bacterial cell envelope compared to colistin treatment alone. To investigate this hypothesis, we employed NPN as a fluorescent probe to assess the outer membrane permeability of *E. coli* ECQ001. By adding DH (8 µg/mL), outer membrane permeability was significantly increased compared to colistin alone (Fig. 3A). However, achieving a comparable increase in permeability required a higher concentration of DH (32 µg/mL), suggesting a dose-dependent enhancement of the activity of colistin by DH (Fig. S1A). Furthermore, inner membrane integrity assays demonstrated that the combination of DH and colistin markedly elevated membrane permeability compared to individual treatments with either compound (Fig. 3B; Fig. S1B). Consistent with these results, the decreased cell membrane stability (Fig. 3C; Fig. S1C) and increased galactosidase release (Fig. 3D; Fig. S1D) in the co-treatment group indicated greater bacterial mortality. Live/dead bacterial staining tests confirmed that DH potentiated the bactericidal efficacy of colistin (Fig. 3E). Collectively, these data supported the notion that DH can enhance colistin-induced damage to both the outer and inner membranes

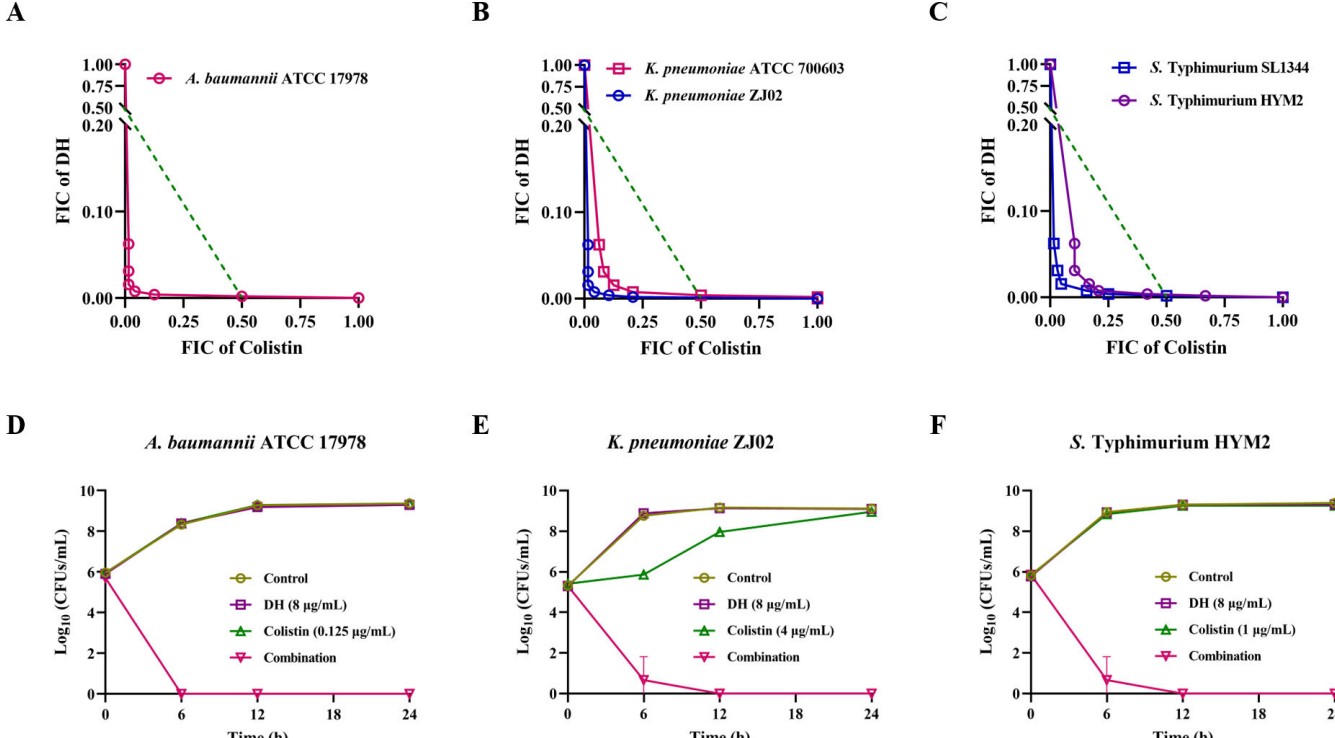

**FIG 2** DH enhances the antibacterial activity of colistin against gram-negative bacteria. (A) Isobolograms of the combination of DH and colistin against *A. baumannii* ATCC 17978. (B) Isobolograms of the combination of DH and colistin against *K. pneumoniae* ATCC 700603 and *K. pneumoniae* ZJ02. (C) Isobolograms of the combination of DH and colistin against *S.* Typhimurium SL1344 and *S.* Typhimurium HYM2. Time-dependent killing curve of *A. baumannii* ATCC 17978 (D), *K. pneumoniae* ZJ02 (E), and *S.* Typhimurium HYM2 (F) treated with DH, colistin, or the combination.

of gram-negative bacteria, leading to increased cellular permeability and ultimately contributing to bacterial cell death. The observed synergy likely stems from a mechanism whereby DH facilitates the penetration and/or action of colistin within the bacterial cell envelope, thereby amplifying its antimicrobial activity.

Bacterial PMF, an energetic pathway located on the bacterial membrane, crucially regulates various biological processes such as adenosine triphosphate synthesis, active transport of molecules, and rotation of bacterial flagella (22). The PMF generally comprises electric potential ($\Delta\Psi$) and transmembrane proton gradient ($\Delta pH$). To investigate the extent of membrane dysfunction induced by DH, we quantified changes in $\Delta\Psi$ and $\Delta pH$ following treatment with DH in combination with colistin. Utilizing the fluorescent probe DiSC$_3$(5), which was sensitive to variations in membrane potential, we observed a significant increase in fluorescence intensity within bacterial cells treated with the combination compared to untreated controls (Fig. 4A; Fig. S1E). This fluorescence enhancement indicated a substantial depolarization of the cytoplasmic membrane. Subsequently, we assessed alterations in $\Delta pH$ using the pH-sensitive fluorescent dye BCECF-AM. Our results demonstrated that the combination groups exhibited more pronounced decreases in $\Delta pH$ relative to the control (Fig. 4B; Fig. S1F). Collectively, these findings suggested that DH potentiated the disruptive effects of colistin on the bacterial cell membrane, leading to significant disruptions in PMF.

Taking into account the observed alteration of membrane integrity caused by DH and colistin, we hypothesized that it also impacted the respiratory chain and energy metabolic pathways within the cytoplasmic membrane (23). To detect this hypothesis, we tested intracellular ATP levels in bacteria exposed to DH and colistin. A marked reduction in ATP production was noted at a DH concentration of 2 µg/mL (Fig. 4C; Fig. S1G), indicating impaired bioenergetic function. The depletion of ATP can disrupt the cellular redox state, leading to the accumulation of reactive oxygen species (ROS)

**TABLE 1** FICI of colistin and DH against different bacterial species[a]

| Strains | Colistin (µg/mL) | | DH (µg/mL) | FICI | Potentiation |
| | Alone | Combination | | | (fold) |
|---|---|---|---|---|---|
| *E. coli* ATCC 25922 | 1 | 1/32 | >256 | 0.05 ± 0.00 | 32 |
| ExPEC ECQ001 | 8 | 1/4 | >256 | 0.05 ± 0.00 | 32 |
| ExPEC 42 | 8 | 1/2 (1) | >256 | 0.10 ± 0.02 | 16 (8) |
| ExPEC 1145 | 4 | 1/2 | >256 | 0.14 ± 0.00 | 8 |
| ExPEC 1209 | 8 | 1/2 | >256 | 0.08 ± 0.00 | 16 |
| *E. coli* B2 | 8 | 1/4 (1/2) | >256 | 0.07 ± 0.01 | 32 (16) |
| *E. coli* DH5α-PME6032 | 1/2 | 1/32 (1/16) | >256 | 0.09 ± 0.01 | 16 (8) |
| *E. coli* DH5α-PME6032 + *mcr-1* | 4 | 1/4 | >256 | 0.08 ± 0.00 | 16 |
| *K. pneumoniae* ATCC 70063 | 1 | 1/16 | >256 | 0.11 ± 0.02 | 16 |
| *K. pneumoniae* ZJ02 | 32 | 1/2 | >256 | 0.03 ± 0.00 | 64 |
| *S.* Typhimurium SL1344 | 2 | 1/16 | >256 | 0.05 ± 0.01 | 32 |
| *S.* Typhimurium HYM2 | 8 | 1/2 (1) | >256 | 0.11 ± 0.03 | 16 (8) |
| *A. baumannii* ATCC 17978 | 1 | 1/64 | >256 | 0.03 ± 0.00 | 64 |
| *S. aureus* Newman | 256 | ND[b] | 8 | ND[b] | ND[b] |

[a]Fractional inhibitory concentration indices (FICIs) were calculated based on chequerboard broth microdilution assays.
[b]ND, not determined.

and the induction of oxidative stress—a key bactericidal mechanism associated with colistin. Consistent with this notion, our data showed a dose-dependent increase in ROS levels as the concentrations of DH and colistin were elevated (Fig. 4D; Fig. S1H). Moreover, we observed a concomitant decrease in superoxide dismutase (SOD) activity (Fig. S2A), suggesting a compromised antioxidant defense system. Taken together, the synergistic bactericidal mechanisms of DH and colistin encompass not only the physical disruption of bacterial membranes but also the impairment of energy metabolism

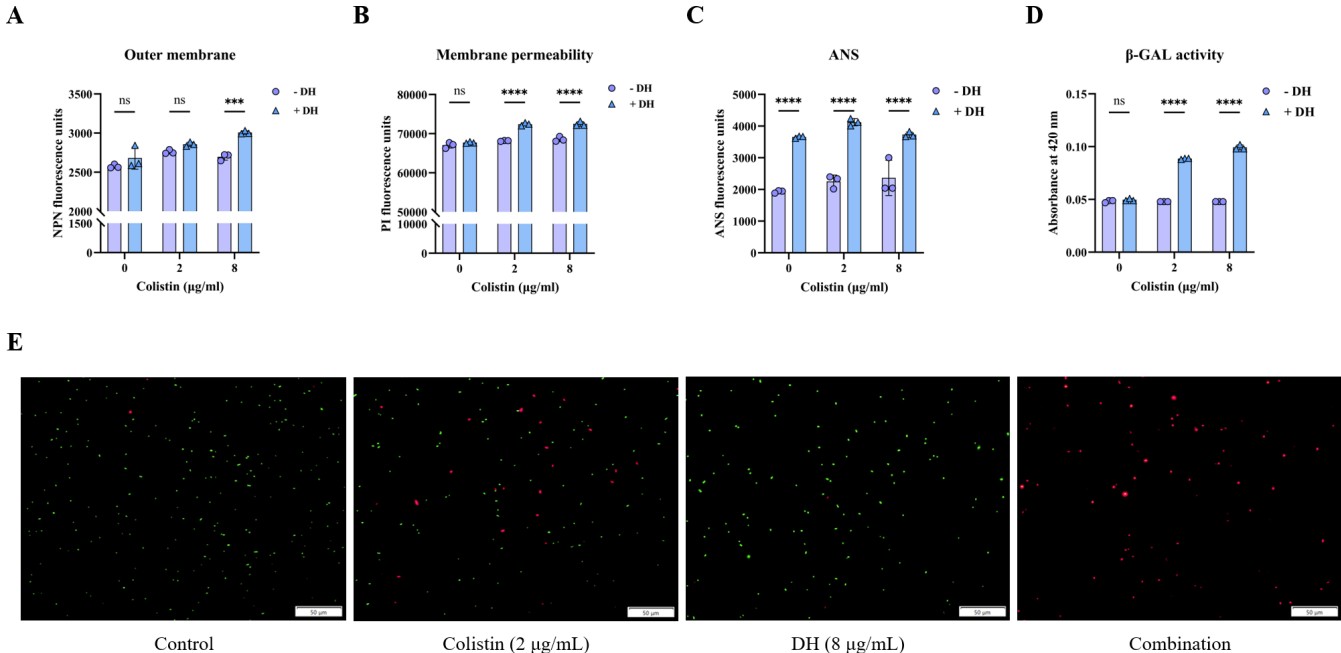

**FIG 3** DH exerts synergy with colistin through disrupting bacterial cell membrane homeostasis. Outer membrane permeability (A), inner membrane permeability (B), and membrane fluidity (C) of *E. coli* ECQ001 treated with DH, colistin, or both. (D) β-galactosidase levels of *E. coli* ECQ001 treated with DH, colistin, or the combination. (E) Live/dead cell images of *E. coli* ECQ001 under the treatment of DH, colistin, or the combination. Red fluorescence represents dead bacteria, and green fluorescence represents live bacteria. Data are expressed as mean ± SD (*n* = 3 per group). Statistical analysis was performed using a two-way ANOVA; ns indicates no significance, ***$P < 0.001$, and ****$P < 0.0001$.

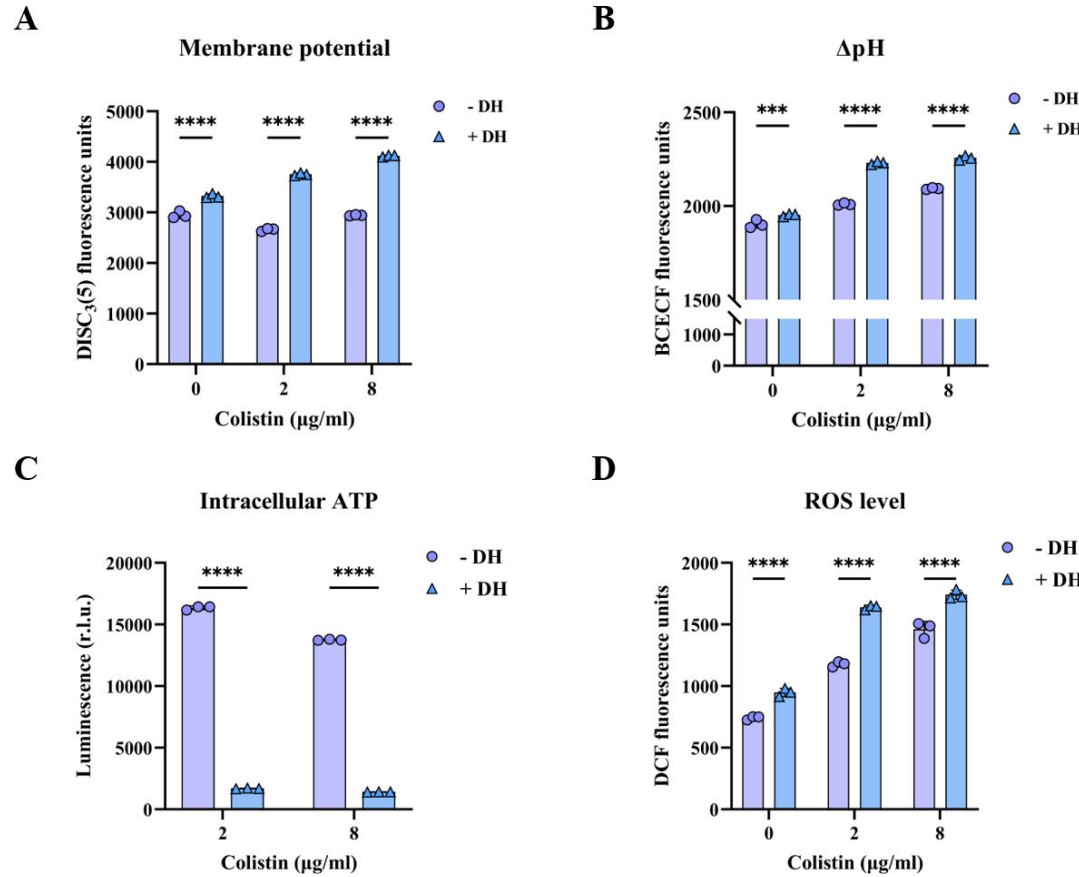

**FIG 4** DH causes bacterial dysfunction. (A) The membrane potential of *E. coli* ECQ001 probed with DiSC$_3$(5). (B) Dissipated ΔpH in *E. coli* ECQ001 treated with DH, colistin, or the combination. (C) The ATP level in *E. coli* ECQ001 treated with DH and colistin. (D) ROS accumulation in *E. coli* ECQ001 treated with DH, colistin, or the combination. Data are presented as mean ± SD ($n$ = 3 per group). Statistical analysis was performed using two-way ANOVA; $^{***}P < 0.001$ and $^{****}P < 0.0001$.

and the induction of oxidative stress. These multifaceted effects collectively contribute to enhanced bacterial lethality, underscoring the potential of DH as an adjuvant for augmenting colistin's therapeutic efficacy against resistant gram-negative pathogens.

## DH targets CL and PG to synergize with colistin

Based on the fact that DH enhanced the disruptive activity of colistin targeting diverse cellular components, we speculated that DH interacted directly with specific components of the outer cell membrane. Phospholipids, which are major components of cell membranes, play crucial roles in numerous biological processes and may serve as potential targets for such interactions. Upon addition of *E. coli* total extract phospholipid profile (PE: 57.5%; CL: 9.8%; PG: 15.1%; and others: 17.6%), we observed a dose-dependent reduction in the synergistic effect of DH and colistin. Specifically, both PG and CL completely abrogated the synergism observed with DH and colistin, whereas PE had no such impact (Fig. 5A). To further elucidate the molecular basis of these observations, we conducted an isothermal titration calorimetry (ITC) assay to quantify the binding affinities between DH and the three phospholipid components. As anticipated, DH exhibited high affinity for PG and CL, with dissociation constants ($K_D$) of $1.3 \times 10^{-6}$ and $2.1 \times 10^{-5}$ M, respectively (Fig. 5C and D). In contrast, DH showed no significant interaction with PE or buffer controls (Fig. S3A and B), reinforcing the specificity of its binding to PG and CL. Notably, our data indicated that the carbon chain length of CL did not influence its binding to DH (Fig. 5B), suggesting that DH preferentially binds to the terminal groups within the glycerol moiety of CL rather than the acyl chains. This selectivity provides insight into the mechanism by which DH enhances colistin's

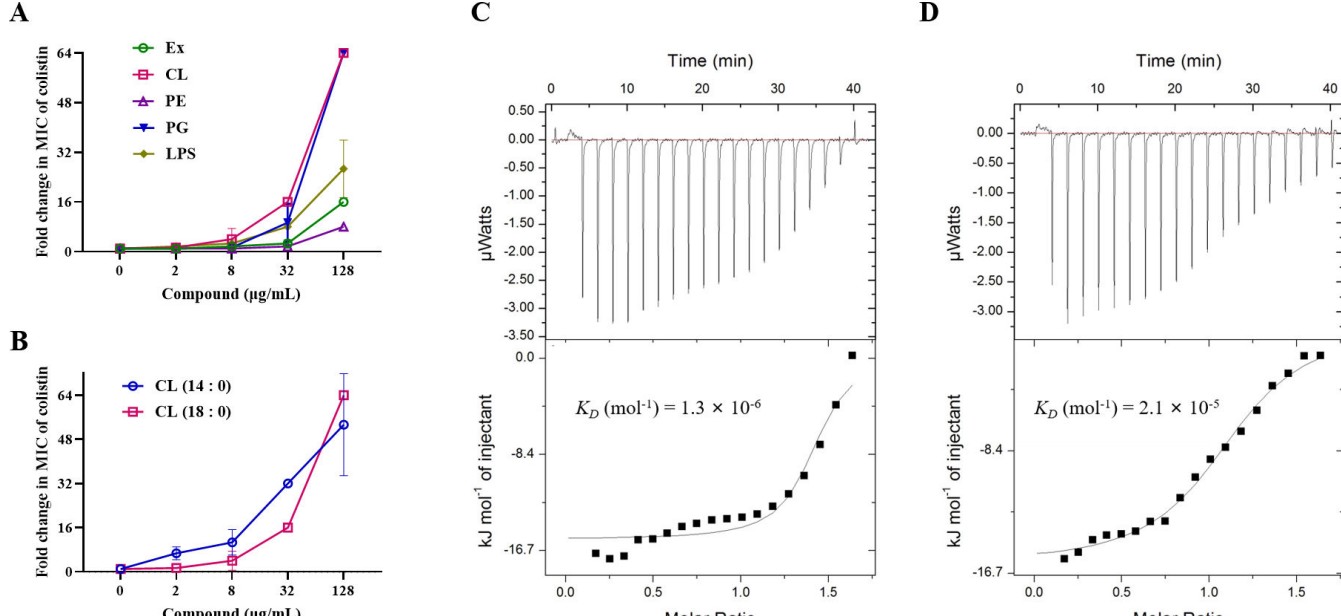

**FIG 5** DH targets CL and PG to synergize with colistin. (A) Changes in colistin MIC against *E. coli* ECQ001 in the presence of exogenous Ex, CL, PE, PG, and LPS. (B) Effect of CL chain length on the MIC of colistin against *E. coli* ECQ001. ITC analysis of binding affinities between DH and PG (C) or CL (D). Data are presented as mean ± SD (*n* = 3 per group).

antibacterial efficacy. In conclusion, our findings demonstrated that DH selectively binds to PG and CL on bacterial membranes, thereby potentiating the disruptive effects of colistin. This interaction appears to be critical for the observed synergistic antibacterial activity of the DH-colistin combination, highlighting the potential of targeting specific phospholipids as a strategy to combat antibiotic-resistant gram-negative pathogens.

## DH potentiates colistin efficacy in a mouse infection model

In light of the remarkable *in vitro* synergistic antimicrobial effects demonstrated by the DH-colistin combination, we proceeded to investigate its therapeutic potential *in vivo*. Before the *in vivo* experiments, we assessed the hemolytic activity of DH to ensure its safety profile. Our results indicated that even at a concentration of 32 µg/mL, DH caused minimal hemolysis of sheep red blood cells (SRBCs) (Fig. S2B), suggesting low cytotoxicity and suitability for further evaluation. To evaluate the efficacy of DH as an antibiotic adjuvant, we employed a mouse model of systemic infection using a clinical isolate of *E. coli* ECQ001. The experimental design is outlined in Fig. 6A. Following treatment, bacterial loads in various organs were significantly reduced in the DH-colistin combination group compared to those treated with colistin or DH alone (Fig. 6B through E). Overall, these results provide a research foundation for the clinical utility of combining DH with colistin, highlighting its potential as a novel adjunctive therapy to address the growing challenge of antibiotic resistance.

## DISCUSSION

Antibiotic adjuvants represent a transformative pharmacological strategy that effectively mitigates antibiotic resistance. Repurposing existing drugs as antibiotic adjuvants offers distinct advantages, such as enhancing the probability of successful research and development endeavors, substantially abbreviating the developmental timeline, and diminishing associated costs (24). Within this study, we conducted a series of bacteriostatic assays to screen for potential adjuvants, ultimately elucidating the capacity of DH in conjunction with colistin to surmount colistin resistance. Our findings indicated that DH decreased the MIC of colistin by approximately 32-fold against colistin-resistant

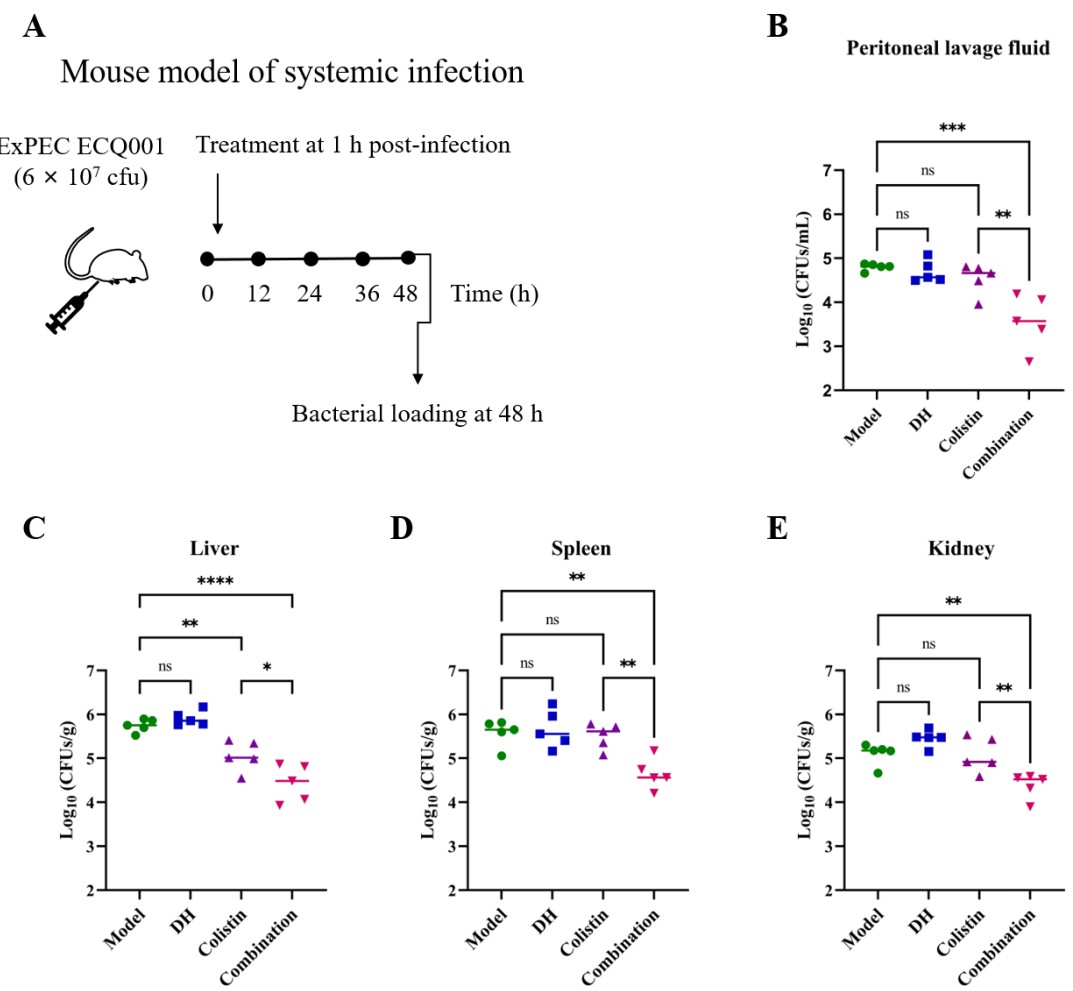

**FIG 6** The synergistic therapeutic efficacy of DH and colistin *in vivo*. (A) Scheme representation of the experimental protocol used in the mouse infection model. Bacterial loads in the peritoneal lavage fluid (B), liver (C), spleen (D), and kidney (E). Data are expressed as mean ± SD (*n* = 5 per group). Statistical analysis was performed using a one-way ANOVA; ns indicates no significance, $^{*}P < 0.05$, $^{**}P < 0.01$, $^{***}P < 0.001$, and $^{****}P < 0.0001$.

ExPEC, thereby potentially mitigating the toxicological profile of colistin while maintaining therapeutic efficacy.

ITC and chequerboard assays suggested that DH bound CL ($K_D = 2.1 \times 10^{-5}$ M) and PG ($K_D = 1.3 \times 10^{-6}$ M) and restored sensitivity to the last-resort antibiotic, colistin. PG and CL are important components of the cell membrane of gram-negative bacteria. Our investigation corroborated previous reports, revealing that compounds targeting these membrane components augment the antimicrobial efficacy of colistin. Given the distinct compositional differences between bacterial and mammalian membranes, selectively targeting bacterial membranes represents a hopeful avenue for developing novel antibacterial agents and antibiotic potentiators.

In the case of *E. coli*, the phospholipid composition of its cytoplasmic membrane predominantly consists of PE (about 75%), PG (approximately 20%), and CL (around 5%), along with minor amounts of other phospholipids such as phosphatidylserine (PS) (25, 26). The selective binding of dehydroabietic acid (DH) to the anionic PG and CL, rather than the more abundant PE, underscores that DH specifically potentiates the antimicrobial efficacy of colistin without exerting direct bactericidal activity against Gram-negative bacteria. In contrast, Gram-positive bacteria exhibited a higher proportion of PG (approximately 50%) and CL (around 30%), with a lower content of PE (27). This phospholipid distribution led us to hypothesize that DH would demonstrate

enhanced antimicrobial activity against gram-positive bacteria, a hypothesis supported by our findings showing a MIC of 8 µg/mL for *Staphylococcus aureus*. Given the minimal presence of PG and CL in mammalian cell membranes (about 1%) (28, 29), the cytotoxic potential of DH towards eukaryotic cells is significantly reduced, thereby highlighting its therapeutic window for antibacterial applications.

Despite being the least abundant among the three major glycerophospholipids in the gram-negative bacterial envelope, CL plays a critical role due to the existence of multiple enzymes (*ClsA*, *ClsB*, and *ClsC*) dedicated to its synthesis in *E. coli* (30, 31). This redundancy underscores the critical importance of CL for bacterial viability and fitness (32). Research has elucidated various biological interactions between CL and essential proteins, including aquaporins, DNA recombination enzymes, and ATP-binding cassette transporters. Moreover, studies by Douglass et al. have shown that CL facilitates the transport of LPS to the outer membrane of gram-negative bacteria, potentially contributing to the synergistic action observed between DH and colistin (33).

In summary, our research demonstrated that DH effectively resensitizes colistin-resistant bacteria to colistin both *in vitro* and *in vivo*. This provided a research foundation for countering the growing threat of colistin-resistant gram-negative bacterial infections.

## MATERIALS AND METHODS

### Chemicals and strains

The bacterial strains used in this study are listed in Table S1. DH and colistin were purchased from APExBIO Technology LLC (Houston, USA) and Shanghai Yuanye Bio-Technology Co., Ltd (Shanghai, China), respectively.

### Mice

Female BALB/c mice aged 6–8 weeks (18–20 g) were obtained from Liaoning Changsheng Biotechnology Co., Ltd. Before infection, mice were acclimatized under standardized environmental conditions, maintaining a temperature of $23 \pm 2°C$ and a humidity level of $55 \pm 10\%$, for a period of 3–5 days. All animal experiments were conducted with the approval of the Animal Care and Use Committee at Jilin University under protocol number SY202412053.

### Chequerboard studies

Fractional inhibitory concentration indexes (FICIs) were determined by chequerboard assays (19). If needed, CL (≥97%, Sigma-Aldrich), PE (≥97%, Avanti-Merk), PG (≥97%, Sigma-Aldrich), or LPS (*E. coli* O111:B4, Sigma-Aldrich) was added to the broth medium to clarify the effect of other substances on the synergy between DH and colistin.

### Growth curves

The effects of DH on bacterial growth were determined by growth curve analysis according to established protocols. In short, the logarithmic growth phase culture was adjusted to $OD_{600}$ nm of 0.2, and DH (0, 8, and 32 µg/mL) was added. The samples were then incubated at 37°C and 180 rpm. Absorbance at 600 nm was measured hourly.

### Time-dependent killing assay

The time-dependent kill curve was carried out to determine the bactericidal effect of DH and colistin as previously described (34).

## Hemolysis assessment

The hemolytic activity of DH was identified according to a previous report (19). Briefly, SRBCs were treated with serial dilutions of DH at 37°C for 1 h. The hemolytic activity of DH was quantified by measuring the absorbance values of the supernatant at $OD_{570\ nm}$.

## Resistance development test

To assess *in vitro* resistance emergence, *E. coli* ECQ001 cultures ($OD_{600\ nm}$ = 0.1) were exposed to colistin (4 µg/mL), DH (8 µg/mL), their combination, or solvent control for 24 h at 37°C with shaking. The same process was repeated for 28 days, and the MIC of colistin for all groups was detected every 3 days.

## ITC assay

ITC experiment was performed using an ITC (TA Instruments) to evaluate the affinity between DH (1 mM, pH 7.0, dissolved in $ddH_2O$ with 3% DMSO) and CL (0.1 mM, pH 7.0, dissolved in $ddH_2O$ with 3% DMSO) or PG (0.1 mM, pH 7.0, dissolved in $ddH_2O$ with 3% DMSO) at 25°C.

## Outer membrane and cell membrane integrity assay

The effect of colistin, DH, or combination on outer membrane permeability of *E. coli* ECQ001 was determined using the fluorescent dyes *N*-phenyl-1-naphthylamine (NPN, AMEKO, China) as previously described (21). Briefly, bacterial suspensions in the logarithmic growth phase were incubated with NPN (10 µM) for 30 min, and then the bacteria were washed two times using sterile PBS. Colistin alone (0, 2, and 8 µg $mL^{-1}$) or in combination with DH (0 and 8 µg $mL^{-1}$) was cultured with bacteria for 1 h. Fluorescence intensity was then measured using a microplate reader (Gen 5, BioTek, USA) at excitation/emission wavelengths of 350 nm/420 nm.

Propidium iodide (PI; 5 µM; Solarbio, China) was used to evaluate inner membrane integrity, with fluorescence measured at excitation/emission wavelengths of 535 nm/615 nm.

## Extracellular β-galactosidase determination

β-galactosidase activity was determined using 2-nitrophenyl-*β*-D-galactopyranoside (ONPG, Yuanye, China) as previously described (19). Logarithmic-phase bacteria were collected, washed, and resuspended in PBS ($OD_{600\ nm}$ = 0.5) and treated with colistin or DH for 3 h at 37°C. After centrifugation (4,500 × *g*, 10 min, 4°C), supernatants were incubated with ONPG (3 mM) for 1 h at 37°C, and absorbance was measured at 420 nm using a microplate reader (Gen 5, BioTek).

## Membrane fluidity test

The bacteria in the logarithmic growth phase were collected, washed, and resuspended in PBS to $OD_{600\ nm}$ = 0.5. And the bacterial suspensions were incubated with 8-anilino-1-naphthalenesulfonic acid ammonium (ANS, 40 µM) for 30 min, and then the bacteria were washed twice using sterile PBS. Colistin alone (0, 2, and 8 µg $mL^{-1}$) or in combination with DH (0, 8 µg $mL^{-1}$) was cultured with bacteria for 1 h. Fluorescence was then measured using a microplate reader (Gen 5, BioTek) at excitation/emission wavelengths of 385 nm/473 nm.

## Live/dead bacteria staining

Overnight *E. coli* ECQ001 cultures were diluted 1:100 in fresh LB medium and grown at 37°C and 220 rpm until $OD_{600\ nm}$ = 0.2–0.3. Bacteria were treated with DH or colistin for 3 h, stained using the live/dead backlight bacterial viability kit (Invitrogen, American) for 15 min, and visualized by fluorescence microscopy (Olympus, Japanese).

## Membrane depolarization assay

As previously reported (19), 3,3-Dipropylthiadicarbocyanine iodide (DisC$_3$(5), Sigma, Germany) was used to measure cell membrane depolarization of *E. coli* ECQ001. Logarithmically growing bacteria were incubated with probes (0.5 µM) for 30 min, then separated into control, colistin-treated (0, 2, and 8 µg/mL), DH-treated (8 µg/mL), and combination groups. After a 1-h incubation, fluorescence was measured at an excitation wavelength of 622 nm and an emission wavelength of 670 nm.

## Δ pH measurement

Changes in transmembrane ΔpH were quantified using 2′,7′-bis-(2-carboxyethyl)-5-(and-6)-carboxyfluorescein (BCECF-AM, Thermo, USA) following established protocols (19). Logarithmically growing bacteria were mixed with BCECF-AM ($2 \times 10^{-6}$ M) and then allocated to control, colistin-treated (0, 2, and 8 µg/mL), DH-treated (8 µg/mL), and combination groups. After a 1-h treatment, fluorescence was measured at an excitation wavelength of 488 nm and an emission wavelength of 535 nm.

## ATP determination

Bacteria in the logarithmic growth phase were harvested in PBS ($OD_{600\ nm} = 0.8$) and then treated with colistin (0, 2, and 8 µg/mL) and DH (8 µg/mL) for 2 h. Intracellular ATP levels were measured using the enhanced ATP assay kit (S0027, Beyotime Biotechnology, China) following the manufacturer's instructions.

## ROS measurement

Bacteria in the logarithmic growth phase were collected in PBS ($OD_{600\ nm} = 0.8$) and then treated with colistin (0, 2, and 8 µg/mL) and DH (8 µg/mL) for 2 h. ROS measurements were performed according to the manufacturer's protocol (Yuanye, China). Oxidative stress-related factors, such as SOD activity, were determined according to the corresponding kit (BC5165, Solarbio, China).

## Mouse infection models

The therapeutic efficacy of DH and colistin was evaluated in a mouse systemic infection model using colistin-resistant *E. coli* ECQ001. After acclimatization feeding, female BALB/c mice were infected intraperitoneally (i.p.) with $6 \times 10^7$ cfu of bacteria suspended in PBS buffer. After 1-h post-infection, mice were treated with a single dose of DH (10 mg/kg, i.p.), colistin (0.25 mg/kg, i.p.) or the combination (DH and colistin). After 48-h post-infection, peritoneal lavage fluid, liver, spleen, and kidney were removed and homogenized in sterile PBS for bacterial loading.

## Statistical analyses

All experiments were performed in at least two independent biological replicates. Data are shown as means ± standard deviation (SD). Statistical analysis and figure generation were performed using GraphPad Prism 9.4.0 (GraphPad Software, USA). Data variability was analyzed using one-way analysis of variance (ANOVA) and two-way ANOVA; [*]indicates $P < 0.05$, [**]indicates $P < 0.01$, [***]indicates $P < 0.001$, [****]indicates $P < 0.0001$, and ns indicates no significance.

## ACKNOWLEDGMENTS

The porcine ExPEC 42, 1209, and 1145 strains, which were isolated from swine cerebrospinal fluid in China, were generously provided by Prof. Chen Tan from Huazhong Agricultural University (35). The *S.* Typhimurium SL1344 strain was kindly donated by Prof. Xiaoyun Liu from Peking University. Additionally, the *E. coli* B2 strain was kindly provided by Prof. Kui Zhu from China Agricultural University.

This study was supported by the National Key Research Development Program of China (2021YFD1800405), the National Natural Science Foundation of China (32373066), the Natural Science Foundation of Jilin Province (20240101282JC), and the Fundamental Research Funds for the Central Universities.

## AUTHOR AFFILIATION

[1]State Key Laboratory for Diagnosis and Treatment of Severe Zoonotic Infectious Diseases, Key Laboratory for Zoonosis Research of the Ministry of Education, College of Veterinary Medicine, Jilin University, Changchun, China

## AUTHOR ORCIDs

Jianfeng Wang http://orcid.org/0000-0001-8311-0894
Hongtao Liu http://orcid.org/0009-0009-2639-2114
Yanhong Deng http://orcid.org/0009-0006-2931-0661
Jiazhang Qiu http://orcid.org/0000-0002-7723-5073

## FUNDING

| Funder | Grant(s) | Author(s) |
| --- | --- | --- |
| National Key Research and Development Program of China | 2021YFD1800405 | Hongtao Liu |
| National Natural Science Foundation of China | 32373066 | Hongtao Liu |
| Natural Science Foundation of Jilin Province | 20240101282JC | Hongtao Liu |
| Fundamental Research Fund for the Central Universities | | Jiazhang Qiu |

## AUTHOR CONTRIBUTIONS

Zhiying Liu, Data curation, Formal analysis, Investigation, Methodology, Supervision, Validation, Visualization, Writing – original draft, Writing – review and editing | Moyun Liu, Data curation, Formal analysis, Investigation, Methodology, Supervision, Validation, Visualization, Writing – review and editing | Zichu Wang, Data curation, Investigation, Supervision, Validation, Visualization, Writing – review and editing | Chenxiao Jiang, Data curation, Investigation, Supervision, Validation, Visualization | Jianfeng Wang, Data curation, Formal analysis, Funding acquisition, Investigation, Project administration, Resources, Supervision, Visualization, Writing – review and editing | Xuming Deng, Conceptualization, Formal analysis, Funding acquisition, Investigation, Project administration, Resources, Supervision, Validation, Writing – review and editing | Hongtao Liu, Conceptualization, Formal analysis, Funding acquisition, Investigation, Methodology, Project administration, Resources, Supervision, Validation, Writing – review and editing | Yanhong Deng, Formal analysis, Funding acquisition, Investigation, Methodology, Project administration, Resources, Supervision, Visualization, Writing – review and editing | Jiazhang Qiu, Conceptualization, Formal analysis, Funding acquisition, Investigation, Methodology, Project administration, Resources, Validation, Visualization, Writing – review and editing

## ADDITIONAL FILES

The following material is available online.

### Supplemental Material

**Supplemental material (Spectrum01196-25-s0001.docx).** Figures S1 to S3; Table S1.

Open Peer Review

**PEER REVIEW HISTORY (review-history.pdf).** An accounting of the reviewer comments and feedback.

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
