## [Reviewer comments · Microbiology Spectrum]

Microbiology Spectrum

Dronedarone hydrochloride targets cardiolipin and phosphatidylglycerol to increase colistin susceptibility in gram-negative pathogens

Zhiying Liu, Moyun Liu, Zichu Wang, Chenxiao Jiang, Jianfeng Wang, Xuming Deng, Hongtao Liu, Yanhong Deng, and Jiazhang Qiu

Corresponding Author(s): Jiazhang Qiu, Jilin University

Review Timeline:

Submission Date:	April 16, 2025
Editorial Decision:	July 7, 2025
Revision Received:	September 2, 2025
Accepted:	September 11, 2025

Editor: Michael Whitfield

Reviewer(s): Disclosure of reviewer identity is with reference to reviewer comments included in decision letter(s). The following individuals involved in review of your submission have agreed to reveal their identity: Kui Zhu (Reviewer #1)

Transaction Report:

DOI: <https://doi.org/10.1128/spectrum.01196-25>

Re: Spectrum01196-25 (Dronedarone hydrochloride targets cardiolipin and phosphatidylglycerol to overcome colistin resistance in gram-negative pathogens)

Dear Prof. Jiazhang Qiu:

Thank you for the privilege of reviewing your work. Below you will find my comments, instructions from the Spectrum editorial office, and the reviewer comments.

Revision Guidelines

Sincerely,
Michael Whitfield
Editor
Microbiology Spectrum

Reviewer #1 (Comments for the Author):

The manuscript reported a potential colistin booster-dronedarone hydrochloride (DH). The authors systematically demonstrate the synergistic antimicrobial effects of DH and colistin through comprehensive in vitro and in vivo assays. By comparing the activity of DH and colistin against mcr-positive and mcr-negative strains, they elucidate its synergistic mechanism. Overall, the study is well-structured, logically presented, and suitable for publication in Microbiology Spectrum. However, the revisions are

recommended to strengthen the manuscript:

Major comments:

1. Although DH was observed to reduce intracellular ATP levels, the primary mechanism underlying its synergistic antimicrobial effect appears to be the induction of oxidative stress. Consequently, the conclusions in the abstract: "The dual targeting of membrane integrity and metabolic homeostasis presents a strategic advantage in circumventing conventional resistance mechanisms, thereby extending the clinical relevance of colistin in contemporary antimicrobial regimens." should be adjusted to align with the full text.
2. This study demonstrates that DH increases membrane permeability, thereby enhancing the antibacterial activity of colistin. Given that heightened membrane permeability may promote the intracellular uptake of other antibiotics, future investigations should explore whether DH exhibits synergistic effects with additional antibiotics.
3. The discussion section indicates that DH exhibits antimicrobial activity against *Staphylococcus aureus*. Does DH exhibit similar antibacterial activity against *E. coli* and other Gram-positive bacteria? Could these findings provide any insights into the synergistic antibacterial mechanisms of DH and colistin?
4. Is the colistin resistance in ECQ001 mediated by chromosomal mutations or a plasmid-borne *mcr* gene? If MCR-independent, does DH influence the expression of phosphatidylethanolamine transferase?

Minor comments:

1. Line 129: Replace short fatty acids (SFAs) with the standard term "short chain fatty acids (SCFAs)" and ensure consistency throughout the text.
2. The structural formula and purity of dronedarone hydrochloride should be presented in the results section.
3. All figure legends (e.g., Fig. S1 and S2) should include statistical analysis methods.
4. Beyond binding phosphatidylglycerol and cardiolipin, does DH exhibit additional synergistic mechanisms with colistin?
5. Consider moving Table S2 to the main text for better accessibility.

Reviewer #2 (Comments for the Author):

This ms investigates dronedarone hydrochloride (DH) as a potentiator to overcome colistin resistance in gram-negative bacteria. Mobile colistin resistance has been drawing much attention; thus the study to address it is of importance. However, the findings from the current study are independent of *mcr* or colistin resistance. The ms is considered not well written to capture the needed information. There is unlikely clinical value of DH for antimicrobial therapy. Major comments are given below

1. A major issue is the lack of some details of the findings and the methodological descriptions that limit the understanding of the investigation (see some examples below).
2. Various descriptions such as promising are considered exaggerated/misled and should be avoided (see some examples below).
3. Title: Can be misleading for colistin-specific resistance but DH's role is independent of colistin resistance determinants. Should change accordingly. (e.g., increase colistin susceptibility).
4. L24-25. It remains to be clinically supported for *mcr*-1-mediated clinical treatment failure. Descriptions should be reworded to avoid misleading.
5. L29-24/L41. Here are merely in vitro activities. Clinical potential for using dronedarone as part of adjunct antimicrobial therapy must be carefully considered. One has to differentiate the issue between the fundamental research and clinical implication.
6. L55. "promising therapeutic strategy" must be avoided at this stage. One has to consider potential clinical regime/safety for dronedarone, a antiarrhythmic medication, to be used in combination with colistin, which itself has significant toxicity. Please do not exaggerate the importance of the findings.
7. L148. One has to understand the intrinsic activities of DH against bacteria. Please provide the MICs of colistin and DH against the strains tested. Table S2 provides FICI results but the FIC determination requires the MICs values of tested substances, including DH. The descriptions are oversimplified. Have the authors also tested other antibiotics besides colistin?
8. L182. The authors only hypothesize and selected envelope as the target. This approach does not rule out other effects.
9. L184. DH at 8 ug/mL was used. It remains unknown for the activity of DH against bacteria such as the MIC values. See later the comments on clinical use of DH for C_{max} level.
10. L186. DH at 32 ug/mL is needed, which is very high and likely argues against the clinical potential for DH. The relevant descriptions for clinical implications must be rewritten.
11. L270/L320. "compelling evidence"/"promising therapeutic strategies" lack adequate support. Please reword.
12. L342-344. Some details should be provided, such as the drugs used and their concentrations.
13. L349. "Safety assessment" is too broad and should be specified with experiment only. The authors could better link the tested doses with the pharmacokinetics of DH used for treating cardiovascular disease in humans. For example, based on a monograph of a DH product, DH repeatedly taken at 400 mg tablet twice daily produced C_{max} of <0.15 ug/mL. Thus, to reach 8 ug/ml, it may not be possible for clinical implication.
14. L353. Resistance development test should be specified for in vitro assay. More importantly, details need to be provided such that others can repeat the study.
15. L395-L409. More details need to be provided for understanding and interpreting the findings.
16. Table S1. Please clearly indicate which isolates are positive regarding colistin resistance and *mcr* status.

17. Table S2. FICI data indicates synergy, regardless of the presence of mcr-1 or not.
18. L14. Corresponding authors: The reviewer can hardly comprehend three corresponding authors for the ms. Note there is only one affiliation for all authors.
19. Minors (examples):
 - L63. No need to have "superbugs"/"formidable" here.
 - L89. Avoid using "CRGNB", which is later used only one.
 - L99/L175. No need to introduce "LPS" twice.
 - L101. Advise replacing "cardinal".
 - L130. Delete "SFAs" which is not used again in the ms.
 - L134. No need to have "AF".L326. "listed", not "list".
 - L330. Add a space after "6-8".
 - L364. Lowercase subheading.
 - L364. Write "an isothermal titration...", not "a Isothermal...". Add "E. coli" before "ECQ001".

There were concerns that the English language usage in the manuscript might make it difficult to properly evaluate the science. The ASM Journals webpage provides links to various language editing services (<https://journals.asm.org/writing-your-paper#language-editing-services>). You may consider using these services when revising your manuscript. The use of these services will have no direct bearing on the editorial decision. ASM has no affiliation with these companies.

Dear editor,

Thank you for handling our manuscript (Spectrum01196-25). We are grateful for the constructive suggestions provided by the reviewers, and the manuscript has been carefully revised according to their comments. The details of the response are as follows. All changes in the manuscript have been highlighted in yellow and are also included in the "Marked Up Manuscript" document.

Reviewer #1 (Comments for the Author):

The manuscript reported a potential colistin booster-dronedarone hydrochloride (DH). The authors systematically demonstrate the synergistic antimicrobial effects of DH and colistin through comprehensive in vitro and in vivo assays. By comparing the activity of DH and colistin against mcr-positive and mcr-negative strains, they elucidate its synergistic mechanism. Overall, the study is well-structured, logically presented, and suitable for publication in Microbiology Spectrum. However, the revisions are recommended to strengthen the manuscript:

Major comments:

1. Although DH was observed to reduce intracellular ATP levels, the primary mechanism underlying its synergistic antimicrobial effect appears to be the induction of oxidative stress. Consequently, the conclusions in the abstract: "The dual targeting of membrane integrity and metabolic homeostasis presents a strategic advantage in circumventing conventional resistance mechanisms, thereby extending the clinical relevance of colistin in contemporary antimicrobial regimens." should be adjusted to align with the full text.

Response: Thanks for your careful review. The synergistic antimicrobial mechanism of DH and colistin is indeed due to elevated oxidation levels leading to bacterial death. We have revised "metabolic homeostasis" to "redox homeostasis". Lines 40-43 in the revised manuscript.

2. This study demonstrates that DH increases membrane permeability, thereby enhancing the antibacterial activity of colistin. Given that heightened membrane permeability may promote the intracellular uptake of other antibiotics, future investigations should explore whether DH exhibits synergistic effects with additional antibiotics.

Response: Thanks for your suggestion. DH enhances the disruptive effect of colistin on cell membranes, yet it does not exhibit significant intrinsic membrane-permeabilizing effects (Figure S1A and S1B). Furthermore, we also assessed the potential synergy between DH and other antibiotic classes; however, no significant antimicrobial synergy was observed.

3. The discussion section indicates that DH exhibits antimicrobial activity against *Staphylococcus aureus*. Does DH exhibit similar antibacterial activity against *E. coli* and other Gram-positive bacteria? Could these findings provide any insights into the synergistic antibacterial mechanisms of DH and colistin?

Response: Thanks for your comments. DH exhibits no antimicrobial activity against Gram-negative bacteria, including *Escherichia coli*, *Salmonella*, and *Klebsiella pneumoniae* (MIC > 256 µg/mL). In contrast, DH demonstrates measurable antimicrobial effects against Gram-positive pathogens, such as *Staphylococcus aureus* and *Clostridium difficile* (MIC = 8 µM) [1]. This marked disparity in antibacterial efficacy between Gram-positive and Gram-negative strains prompted us to hypothesize that DH synergizes with colistin by targeting the membrane components. And it was also validated in this manuscript. The corresponding discussion section is in Lines 305-308.

4. Is the colistin resistance in ECQ001 mediated by chromosomal mutations or a plasmid-borne mcr gene? If MCR-independent, does DH influence the expression of

phosphatidylethanolamine transferase?

Response: Thank you for your suggestion. *E. coli* ECQ001 is chromosomally mediated colistin resistance without the *mcr* gene. Therefore, we did not assess the effect of DH on phosphatidylethanolamine transferase expression.

Minor comments:

1. Line 129: Replace short fatty acids (SFAs) with the standard term "short chain fatty acids (SCFAs)" and ensure consistency throughout the text.

Response: Thanks for your careful review. We have revised it to the correct abbreviation and checked for writing errors throughout the manuscript. Lines 131-132 in the revised manuscript.

2. The structural formula and purity of dronedarone hydrochloride should be presented in the results section.

Response: Thanks for your suggestion. We have supplemented the structural formula and purity of DH (99.77%) in the Results section. Please see the revised Figure S2C.

3. All figure legends (e.g., Fig. S1 and S2) should include statistical analysis methods.

Response: Thanks for your suggestion. We have added the statistical analysis methods in the figure legends.

4. Beyond binding phosphatidylglycerol and cardiolipin, does DH exhibit additional synergistic mechanisms with colistin?

Response: Thanks for your suggestion. Our research shows that DH works together with colistin by targeting bacterial membrane phospholipids-a key mechanism we confirmed through experiments. We also found that DH disrupts bacterial redox homeostasis and energy production, which we linked to the phospholipid pathway. The biological effects of compounds are always varied. Although additional synergistic mechanisms between DH and colistin may exist, the targets identified in this study have been verified.

5. Consider moving Table S2 to the main text for better accessibility.

Response: Thanks for your careful review. Table S2 has been moved to the main text. Please see the Table 1 in the revised manuscript.

Reviewer #2 (Comments for the Author):

This ms investigates dronedarone hydrochloride (DH) as a potentiator to overcome colistin resistance in gram-negative bacteria. Mobile colistin resistance has been drawing much attention; thus the study to address it is of importance. However, the findings from the current study are independent of mcr or colistin resistance. The ms is considered not well written to capture the needed information. There is unlikely clinical value of DH for antimicrobial therapy. Major comments are given below.

1. A major issue is the lack of some details of the findings and the methodological descriptions that limit the understanding of the investigation (see some examples below).

Response: Thanks for your suggestion. We have supplemented the details in the Methods section.

2. Various descriptions such as promising are considered exaggerated/misled and should be avoided (see some examples below).

Response: Thanks for your suggestion. We have revised these descriptions into more appropriate expressions.

3. Title: Can be misleading for colistin-specific resistance but DH's role is independent of colistin resistance determinants. Should change accordingly. (e.g., increase colistin susceptibility).

Response: Thank you for your careful review and constructive suggestion. We have revised the title to “Dronedarone hydrochloride targets cardiolipin and phosphatidylglycerol to increase colistin susceptibility in gram-negative pathogens”.

4. L24-25. It remains to be clinically supported for mcr-1-mediated clinical treatment failure. Descriptions should be reworded to avoid misleading.

Response: Thanks for your suggestion. We have polished the sentence in Lines 21-25. The details are as follows: This reliance has precipitated a concerning epidemiological trend: the emergence and global propagation of plasmid-mediated (*mcr*) as well as chromosome-mediated polymyxin resistance. Consequently, escalating resistance rates will certainly lead to diminished clinical efficacy of colistin, correlating with elevated mortality in septic patients who already face therapeutic limitations. Many studies have reported cases of colistin therapy failure [2-4].

5. L29-24/L41. Here are merely *in vitro* activities. Clinical potential for using dronedarone as part of adjunct antimicrobial therapy must be carefully considered. One has to differentiate the issue between the fundamental research and clinical implication.

Response: Thanks for your comment. We actually performed an *in vivo* experimental therapeutic study using a murine infection model. The data showed that compared to colistin therapy alone, combination therapy significantly reduced the bacterial load in the liver, spleen, kidneys, and peritoneal cavity. We have also added relevant findings in the Abstract section in Lines 31-33. Despite this, we agree with you that this is fundamental research. More research is required before clinical application in synergistic therapy with colistin. In addition, we have polished the corresponding conclusions. Lines 40-43 in the revised manuscript.

6. L55. "promising therapeutic strategy" must be avoided at this stage. One has to consider potential clinical regime/safety for dronedarone, a antiarrhythmic medication, to be used in combination with colistin, which itself has significant toxicity. Please do not exaggerate the importance of the findings.

Response: Thanks for your suggestion. The description "promising therapeutic strategy" is indeed inaccurate, and we have revised it to "lead compound" in Lines 56-58. It is widely believed that colistin has significant nephrotoxicity and neurotoxicity. But there are a few cases of colistin causing neurological damage, while the incidence of nephrotoxicity

is relatively high (20%-50%) [5, 6]. In clinical treatment, nephrotoxicity caused by colistin often occurs in elderly patients or those using other nephrotoxic drugs. When nephrotoxicity occurs during colistin treatment, doctors need to adjust the treatment regimen to minimize the nephrotoxicity associated with colistin. In addition, colistin is often used in combination with other antibiotics to lower its dosage and nephrotoxicity. For example, when treating Gram-negative drug-resistant bacterial lung infections, nebulized colistin together with intravenous administration of antibiotics with lower nephrotoxicity can be used. Our goal is also to reduce the dosage of colistin as much as possible [7].

7. L148. One has to understand the intrinsic activities of DH against bacteria. Please provide the MICs of colistin and DH against the strains tested. Table S2 provides FICI results but the FIC determination requires the MICs values of tested substances, including DH. The descriptions are oversimplified. Have the authors also tested other antibiotics besides colistin?

Response: Thanks for your constructive comments. We have supplemented the MICs of colistin and DH against the strains tested. (Please see the Table 1 in the revised manuscript). Additionally, we did synergistic antimicrobial effects between DH and other types of antibiotics such as meropenem, tetracycline, and gentamicin. However, our data revealed no evidence of synergistic activity with these combinations. Following your suggestion, we have improved the description of screening for colistin booster in Lines 151-155.

8. L182. The authors only hypothesize and selected envelope as the target. This approach does not rule out other effects.

Response: Thanks for your suggestion. The biological effects of compounds are always multiple. A pertinent example is metformin, which is not only the first-line drug for type 2 diabetes mellitus (T2DM) but also exhibits ancillary benefits such as weight reduction and, as recently reported, tetracycline synergy [8, 9]. Conversely, it also carries adverse effects, including nausea and diarrhea.

Regarding DH, our mechanistic investigations, grounded in experimental evidence,

demonstrated that membrane phospholipids play a pivotal role in mediating the synergy between DH and colistin. Furthermore, based on the phospholipid pathway, we postulated and subsequently validated that DH perturbs bacterial redox homeostasis and energy metabolism. While additional synergistic mechanisms between DH and colistin may exist, the targets identified in this study have been substantiated.

We have not found any relevant clues about the synergistic mechanism of DH and colistin in PubMed. To systematically study the synergistic mechanism of DH and colistin, we may need to utilize tools such as transcriptomics and proteomics. The possible mechanism will be identified through transcriptomics and further investigated using appropriate research methods.

9. L184. DH at 8 ug/mL was used. It remains unknown for the activity of DH against bacteria such as the MIC values. See later the comments on clinical use of DH for Cmax level.

Response: Thanks for your careful review. It is indeed our mistake not to label the MICs of DH against bacteria, which we have added in Table S2 (Please see the Table 1 in the revised manuscript). As demonstrated in Figure 1A, the MIC of DH against *E. coli* ECQ001 exceeds 128 µg/mL. Furthermore, the growth curve analysis presented in Figure 1B confirms that DH at 32 µg/mL exhibits no detectable antibacterial activity, as evidenced by the absence of any significant growth inhibition compared to the untreated control.

10. L186. DH at 32 ug/mL is needed, which is very high and likely argues against the clinical potential for DH. The relevant descriptions for clinical implications must be rewritten.

Response: Thanks for your suggestion. We apologize that our description has caused you to misunderstand. When used synergistically with colistin, the required DH concentration is 8 µg/mL. Lines 187-189 in the revised manuscript. According to our FIC results, DH at 2 µg/mL can reduce the MIC of colistin by 2-8 times, while DH at 4 µg/mL can reduce the MIC by 2-16 times. Lower doses of DH still exhibit a synergistic effect with colistin, but the best synergistic effect with colistin is 8 µg/mL. For example, the

combination shows excellent antimicrobial activity against *Acinetobacter baumannii* and *Klebsiella pneumoniae*, and other strains are detailed in Table 1.

Both *Acinetobacter baumannii* and *Klebsiella pneumoniae* are common causative agents of bacterial pneumonia. For carbapenem-resistant bacterial pneumonia, colistin is usually the antibiotic of last resort, but the risk of therapeutic failure is also higher [10]. Whether DH can be utilized as a colistin potentiator does require a lot of research, and we only intended to serve as a lead compound as well as a reference for basic and clinical research.

11. L270/L320. "compelling evidence"/"promising therapeutic strategies" lack adequate support. Please reword.

Response: Thanks for your suggestion. We have revised this to more precise scientific terminology, e.g., provide research foundation.

Lines 273-275 in the revised manuscript: Overall, these results provide research foundation for the clinical utility of combining DH with colistin, highlighting its potential as a novel adjunctive therapy to address the growing challenge of antibiotic resistance.

Lines 322-324 in the revised manuscript: This provided research foundation for counteracting the growing threat of colistin-resistant Gram-negative bacterial infections.

12. L342-344. Some details should be provided, such as the drugs used and their concentrations.

Response: Thanks for your suggestion. We have added the concentrations in the

Methods section. The specific description is as follows in Lines 346-349: Effects of DH on bacterial growth were determined by growth curve analysis according to established protocols. In short, the logarithmic growth phase culture was adjusted to OD_{600 nm} of 0.2, and DH (0, 8, and 32 µg/mL) was added. The samples were then incubated at 37°C and 180 rpm. Absorbance at 600 nm was measured hourly.

13. L349. "Safety assessment" is too broad and should be specified with experiment only. The authors could better link the tested doses with the pharmacokinetics of DH used for treating cardiovascular disease in humans. For example, based on a monograph of a DH product, DH repeatedly taken at 400 mg tablet twice daily produced C_{max} of <0.15 µg/mL. Thus, to reach 8 µg/ml, it may not be possible for clinical implication.

Response: Thanks for your careful review. We have revised the "Safety assessment" to "Hemolysis assessment" in Line 353.

The oral bioavailability of DH is approximately 4%, which increases to ~15% when co-administered with a high-fat meal [11]. Ranpise et al. demonstrated that DH loaded in solid lipid nanoparticles exhibited a 2.68-fold enhancement in bioavailability, achieving a plasma concentration of 0.5 µg/mL (C_{max} = 480.95 ± 20.62 ng/mL) [11]. Pharmacokinetic studies in rat models revealed that an oral dose of 55 mg/kg DH yielded plasma concentrations of 1.13 µg/mL in healthy rats, while hyperlipidemic rats showed significantly higher concentrations (3.27 µg/mL) [12].

Given these pharmacokinetic limitations of oral administration, alternative delivery methods warrant consideration. Intravenous administration could potentially achieve higher systemic concentrations. Alternatively, for targeted treatment of respiratory infections such as *Klebsiella pneumoniae*-induced pneumonia, pulmonary delivery via nebulization may offer therapeutic advantages. Notably, our murine infection models have demonstrated synergistic therapeutic efficacy between DH and colistin, further supporting its potential.

14. L353. Resistance development test should be specified for in vitro assay. More importantly, details need to be provided such that others can repeat the study.

Response: Thanks for your suggestion. We have clearly labeled it as an *in vitro* resistance development test. And, the details of the Resistance development test have been added to the Methods section in Lines 358-362. The detailed description is as follows: To assess *in vitro* resistance emergence, *E. coli* ECQ001 cultures ($OD_{600\text{ nm}} = 0.1$) were exposed to colistin (4 $\mu\text{g/mL}$), DH (8 $\mu\text{g/mL}$), their combination, or solvent control for 24 h at 37 °C with shaking. The same process was repeated for 28 days, and the MIC of colistin for all groups was detected every 3 days.

15. L395-L409. More details need to be provided for understanding and interpreting the findings.

Response: Thanks for your suggestion. We have added details in the Methods section. The details are in Lines 400-426.

16. Table S1. Please clearly indicate which isolates are positive regarding colistin resistance and *mcr* status.

Response: Thanks for your suggestion. We have labeled all strains to reflect both their colistin resistance phenotype and *mcr* carriage status in Table S1.

17. Table S2. FICI data indicates synergy, regardless of the presence of *mcr-1* or not.

Response: According to our results, it is. Furthermore, based on this phenomenon, we speculate that the synergistic mechanism of DH and colistin does not depend on MCR. In addition to plasmid-mediated resistance, chromosome-mediated resistance to colistin is also a serious problem. Therefore, the compounds discovered in this article have a broader range of synergistic antibacterial activity with colistin.

18. L14. Corresponding authors: The reviewer can hardly comprehend three corresponding authors for the ms. Note there is only one affiliation for all authors.

Response: Yes, we're all in the same institution. We found that Microbiology Spectrum also had papers with three corresponding authors recently [13, 14].

19. Minors (examples):

L63. No need to have "superbugs"/"formidable" here.

L89. Avoid using "CRGNB", which is later used only one.

L99/L175. No need to introduce "LPS" twice.

L101. Advise replacing "cardinal".

L130. Delete "SFAs" which is not used again in the ms.

L134. No need to have "AF".L326. "listed", not "list".

L330. Add a space after "6-8".

L364. Lowercase subheading.

L364. Write "an isothermal titration...", not "a Isothermal...". Add "E. coli" before "ECQ001".

Response: Thanks for your careful review. All these issues have been revised in the corresponding locations.

Reference

1. Abouelkhair, A.A. and M.N. Seleem, *Exploring novel microbial metabolites and drugs for inhibiting Clostridioides difficile*. mSphere, 2024: p. e0027324.
2. Band, V.I., et al., *Carbapenem-Resistant Klebsiella pneumoniae Exhibiting Clinically Undetected Colistin Heteroresistance Leads to Treatment Failure in a Murine Model of Infection*. mBio, 2018. **9**(2).
3. Paul, M., et al., *Colistin alone versus colistin plus meropenem for treatment of severe infections caused by carbapenem-resistant Gram-negative bacteria: an open-label, randomised controlled trial*. Lancet Infect Dis, 2018. **18**(4): p. 391-400.
4. Kaye, K.S., et al., *Efficacy and safety of sulbactam-durlobactam versus colistin for the treatment of patients with serious infections caused by Acinetobacter baumannii-calcoaceticus complex: a multicentre, randomised, active-controlled, phase 3, non-inferiority clinical trial (ATTACK)*. Lancet Infect Dis, 2023. **23**(9): p. 1072-1084.
5. Spapen, H., et al., *Renal and neurological side effects of colistin in critically ill patients*. Ann Intensive Care, 2011. **1**(1): p. 14.
6. Kaye, K.S., et al., *Colistin Monotherapy versus Combination Therapy for Carbapenem-Resistant Organisms*. NEJM Evid, 2023. **2**(1): p. 103742.
7. Tsuji, B.T., et al., *International Consensus Guidelines for the Optimal Use of the Polymyxins: Endorsed by the American College of Clinical Pharmacy (ACCP), European Society of Clinical Microbiology and Infectious Diseases (ESCMID), Infectious Diseases Society of America (IDSA), International Society for Anti-infective Pharmacology (ISAP), Society of Critical Care Medicine (SCCM), and Society of Infectious Diseases Pharmacists (SIDP)*. Pharmacotherapy, 2019. **39**(1): p. 10-39.
8. Coll, A.P., et al., *GDF15 mediates the effects of metformin on body weight and energy balance*. Nature, 2020. **578**(7795): p. 444-448.
9. Liu, Y., et al., *Metformin Restores Tetracyclines Susceptibility against Multidrug Resistant Bacteria*. Adv Sci (Weinh), 2020. **7**(12): p. 1902227.
10. Buendía, J.A., D. Guerrero Patiño, and A.F. Zuluaga Salazar, *Efficacy of adjunctive inhaled colistin and tobramycin for ventilator-associated pneumonia: systematic review and meta-analysis*. BMC Pulm Med, 2024. **24**(1): p. 213.
11. Gambhire, V.M., M.S. Gambhire, and N.S. Ranpise, *Solid Lipid Nanoparticles of Dronedarone Hydrochloride for Oral Delivery: Optimization, In Vivo Pharmacokinetics and Uptake Studies*. Pharm Nanotechnol, 2019. **7**(5): p. 375-388.
12. Jardan, Y.A. and D.R. Brocks, *The pharmacokinetics of dronedarone in normolipidemic and hyperlipidemic rats*. Biopharm Drug Dispos, 2016. **37**(6): p. 345-51.
13. Shiqing, Y., et al., *Clostridium butyricum enhances cognitive function in APP/PS1 mice by modulating neuropathology and regulating acetic acid levels in the gut microbiota*. Microbiol Spectr, 2025: p. e0017825.
14. Phan, J., et al., *Gut health predictive indices linking gut microbiota dysbiosis with healthy state, mild gut discomfort, and inflammatory bowel disease phenotypes using gut microbiome profiling*. Microbiol Spectr, 2025: p. e0027125.

Re: Spectrum01196-25R1 (Dronedarone hydrochloride targets cardiolipin and phosphatidylglycerol to increase colistin susceptibility in gram-negative pathogens)

Dear Prof. Jiazhang Qiu:

Your manuscript has been accepted, and I am forwarding it to the ASM production staff for publication. Your paper will first be checked to make sure all elements meet the technical requirements. ASM staff will contact you if anything needs to be revised before copyediting and production can begin. Otherwise, you will be notified when your proofs are ready to be viewed.

Sincerely,
Michael Whitfield
Editor
Microbiology Spectrum